# Publication bias in psychology: A closer look at the correlation between sample size and effect size

**Audrey Helen Linden**[1,2], **Thomas V. Pollet**[3], **Johannes Hönekopp**[3]*

**1** Centre for Research in Autism and Education (CRAE) Department of Psychology, University College London, London, United Kingdom, **2** Department of Psychology and Counselling, The Open University, Walton, United Kingdom, **3** Department of Psychology, Northumbria University, Northumbria, United Kingdom

* johannes.honekopp@unn.ac.uk

**Data Availability Statement:** The data and analysis document can be found at https://osf.io/ce6v3/?view_only=86b6b997ca52430898a6a2bdb38cf9bb.

## Abstract

Previously observed negative correlations between sample size and effect size (n-ES correlation) in psychological research have been interpreted as evidence for publication bias and related undesirable biases. Here, we present two studies aimed at better understanding to what extent negative n-ES correlations reflect such biases or might be explained by unproblematic adjustments of sample size to expected effect sizes. In Study 1, we analysed n-ES correlations in 150 meta-analyses from cognitive, organizational, and social psychology and in 57 multiple replications, which are free from relevant biases. In Study 2, we used a random sample of 160 psychology papers to compare the n-ES correlation for effects that are central to these papers and effects selected at random from these papers. n-ES correlations proved inconspicuous in meta-analyses. In line with previous research, they do not suggest that publication bias and related biases have a strong impact on meta-analyses in psychology. A much higher n-ES correlation emerged for publications' focal effects. To what extent this should be attributed to publication bias and related biases remains unclear.

## Introduction

A spectre is haunting psychology–the spectre of bias. The erroneous belief that statistically non-significant findings are uninformative incentivises researchers to publish statistically significant findings [1,2]. As a consequence, researchers might selectively report those analyses and outcomes that turn out statistically significant, and they might keep their statistically non-significant studies in the file drawer [3,4]. These biases, collectively known as publication selection bias (PSB), cause a problematic inflation of favourable evidence in the published literature [5]. As a consequence, treatments might be less effective than believed and PhD students and researchers might waste their time investigating imaginary effects.

PSB is a concern across many disciplines [6,7]. Multiple lines of evidence indicate that psychology is affected too. Thus, effect sizes are often substantially smaller: in unpublished than in published studies [8,9]; in replications than in the original studies being replicated [10]; and in

**Funding:** The authors received no specific funding for this work.

**Competing interests:** The authors have declared that no competing interests exist.

studies with some pre-registration of hypotheses, data collection methods, and analyses (all of which limit PSB) than in studies without pre-registration [11]. Also, registered reports (which preclude PSB because details of data collection, reporting standards, and publication are agreed before the study commences) find evidence in favour of their central hypothesis much less frequently than conventional studies do [12]. Moreover, PSB is often suggested by various techniques developed to detect its prevalence in meta-analyses [8,13].

In light of this evidence for PSB and its negative consequences in psychology, an indicator would be desirable to reflect how serious the problem is and, perhaps more importantly, whether PSB reduces over time, e.g., due to the effectiveness of proposed counter-measures such as study pre-registration [14]. As we shall describe in greater detail below, the indicators that we have discussed so far (e.g., comparison of results across studies that are more or less prone to PSB) show serious limitations for these purposes. More suitable towards these ends might be the correlation between studies' sample size $n$ and their effect size (henceforth n-ES correlation), as we argue in detail below. Here, we present two studies that aim to better understand the validity of the n-ES correlation as an indicator of PSB in psychology.

Measures that proved valuable for flagging PSB as a potential problem might be less suitable to indicate how widespread the problem is or how PSB changes (or fails to change) over time. Effect size comparisons between published and unpublished studies are hampered by the fact that the latter are difficult to obtain without bias [8]. Effect size comparisons between original studies and their replications are limited by the relatively small number of replications and the lack of representativeness in the studies chosen for replication (e.g., a short online study that shows astonishing results will be more likely to be replicated than an arduous clinical trial that finds a small effect). Given the relative novelty of pre-registered studies and registered reports, analyses reliant on them have limited value for studying change over time. Finally, techniques that have been developed to uncover PSB within meta-analyses often disagree in their conclusions, suffer from low statistical power, and generally struggle in the face of effect size heterogeneity (i.e., when the true magnitude of the effect under investigation varies across studies), which is almost ubiquitous [15–17].

The n-ES correlation is an alternative indicator of PSB, which avoids these problems. Its logic is best illustrated when we imagine a set of studies that investigate the same effect in the same way but differ in their $n$s. Let us assume that all studies with $p < .05$ get published and all studies with $p \geq .05$ get rejected. Across studies, the observed effect sizes fluctuate symmetrically around the true population effect size (with this fluctuation being stronger in smaller studies than in larger studies, see funnel plot in Fig 1). Whether a study's $p$-value turns out low enough to result in publication hinges on two factors: the study's $n$ (ceteris paribus, larger $n$s result in smaller $p$-values) and the study's observed effect size (ceteris paribus, larger observed effect sizes result in smaller $p$-values). Consequently, the threshold for the smallest observed effect size that satisfies $p < .05$ decreases as $n$ increases; in the subgroup of published studies, a negative n-ES correlation therefore emerges, which is absent in the complete set of studies (see Fig 1). Consequently, a negative n-ES correlation might indicate PSB. (Our example only considered publication being contingent on statistical significance, whereas PSB encompasses additional biases such as selective reporting of outcomes and analyses [4]. These additional biases, however, contribute to a negative n-ES correlation for similar reasons.)

The n-ES correlation avoids the problems discussed for other PSB indicators above. This can be illustrated by an influential survey that looked at a random sample of almost 400 psychology papers from 2007 and found a strong negative n-ES correlation [18]. Hereafter, we refer to samples of this type that compile data across a wide range of topics as *cross-topics samples*. For cross-topics samples, statistical power is not a concern because researchers can compile as large a sample of studies as is required. Also, random sampling of studies, which

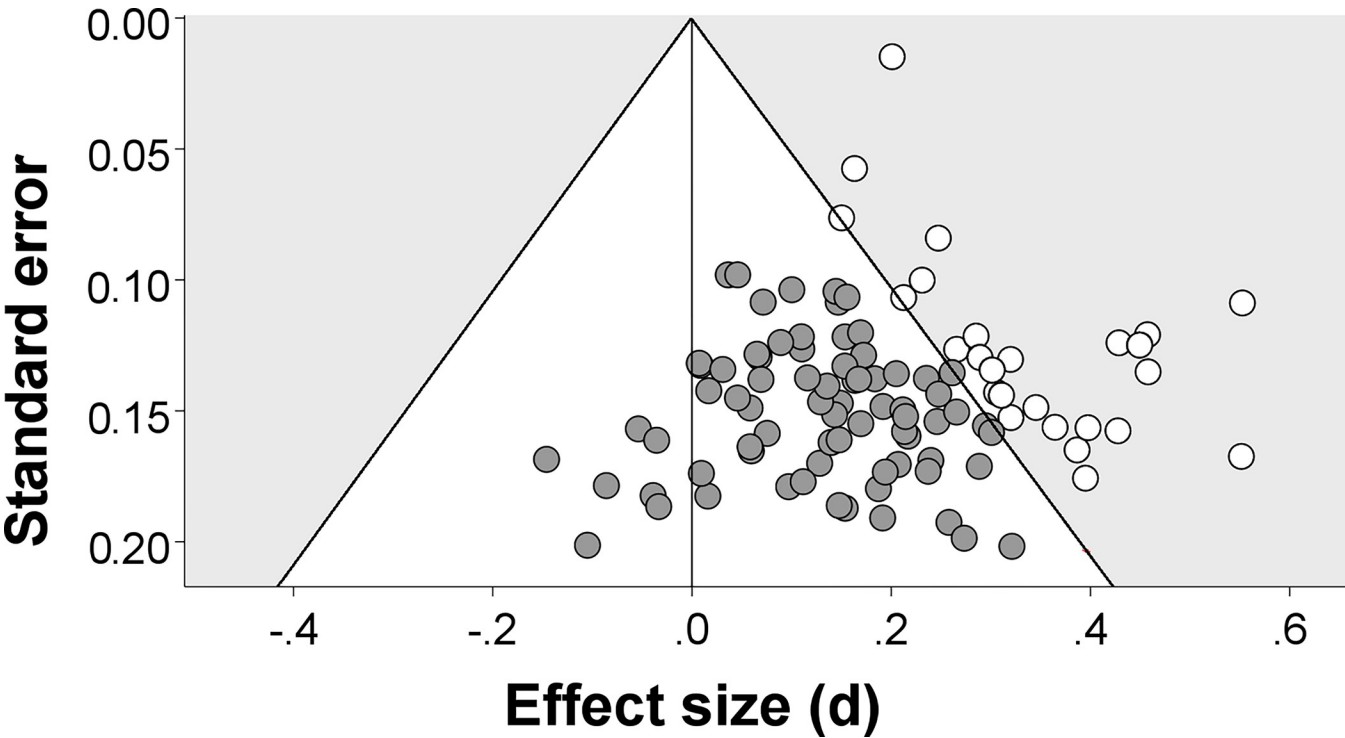

**Fig 1. Observed effect sizes for 100 simulated studies scatter symmetrically around the true population effect size *d* = 0.2.** Studies with larger *N* have smaller standard errors and are therefore located towards the top. Only study results within the grey area are statistically significant ($p < .05$). If only these studies (white) are published, a correlation between sample size and effect size emerges (here, $r = -.59$), which is absent for the complete set of studies.

guarantees the representativeness of the sample for the target population, is easy to achieve. Effect size heterogeneity, however, remains a potential problem in cross-topics samples because innocuous factors other than PSB can lead to a negative n-ES correlation. A particular concern is that researchers have some understanding of the magnitude of the effect they are studying and adjust their sample size accordingly, i.e., use larger samples to study small effects and smaller samples to study large effects. This will lead to a negative n-ES correlation even in the absence of PSB, and we shall refer to this as sample-size adjustment. Although, sample size calculations are infrequent in psychology [18], this does not invalidate concerns over sample-size adjustment because researchers might have tacit knowledge about which *n* suffices in their field. Consequently, it is unclear to what extent the strong negative n-ES correlation found by [18] represents PSB or sample-size adjustment. In Study 1, we sought to explore this issue. We did so by analysing the n-ES correlation under a range of circumstances that differ in key aspects. Given the complexity of the issue, some readers might find Table 2 a helpful companion to the detailed account that follows below.

### Aims

The first aim of Study 1 was to investigate the n-ES correlation under circumstances that make sample-size adjustment unlikely. We did this by computing the n-ES correlation for the studies combined within the same meta-analysis (henceforth, *within meta-analyses*) E.g., this would be $r = -.59$ for a fictitious meta-analysis of the white studies in Fig 1. What drives the difference in effect sizes among studies that are combined in a meta-analysis, remains typically unclear [15,19]. This suggests that researchers are unable to predict if their effect will turn out small,

average, or large compared to other studies of the same topic. Consequently, diverse investigations of the same topic should be based on the same expectation of effect size, and this should largely eliminate sample-size adjustment within meta-analyses. (There are further reasons why a negative n-ES correlation might arise in the absence of PSB [20]. However, these reasons concern specific characteristics of medical trials that rarely apply to psychological research, which is why we do not pursue this point further.)

In order to judge if the observed (average) r-ES correlation within meta-analyses is indicative of PSB, it is important to know what to expected in the absence of PSB. Intuitively, $r = .00$ appears correct. However, even without PSB, negative n-ES correlations might arise (see Fig 2).

Consequently, it would be helpful to compare the n-ES correlation within-meta-analyses (where PBS might be an issue) against data that are free from PBS. Many-Labs replications and Registered Replication Reports (hereafter *multiple replications*) present such an opportunity. Multiple replications use standardised procedures to replicate original studies across multiple sites (e.g., [21,22]. Because any set of replications addresses the same original study, sample-size adjustment cannot be an issue. Additionally, because multiple replications are pre-registered, PSB can be expected to be absent, too. We therefore determined the n-ES correlation within each set of multiple replications to obtain a PSB-free comparison standard for the n-ES correlation observed within meta-analyses. If we were to find a stronger n-ES correlation within-meta-analyses than within multiple replications, this would suggest PSB within meta-analyses. Conversely, if the (average) n-ES correlation turned out to be similar within meta-analyses and within multiple-replications, this would suggest the absence of PSB in meta-analyses.

The second aim of Study 1 was to explore evidence for sample-size adjustment across topics and its impact on the n-ES correlation in cross-topics samples. Plausibly, researchers use relatively small samples to investigate topics that typically produce strong effects and relatively

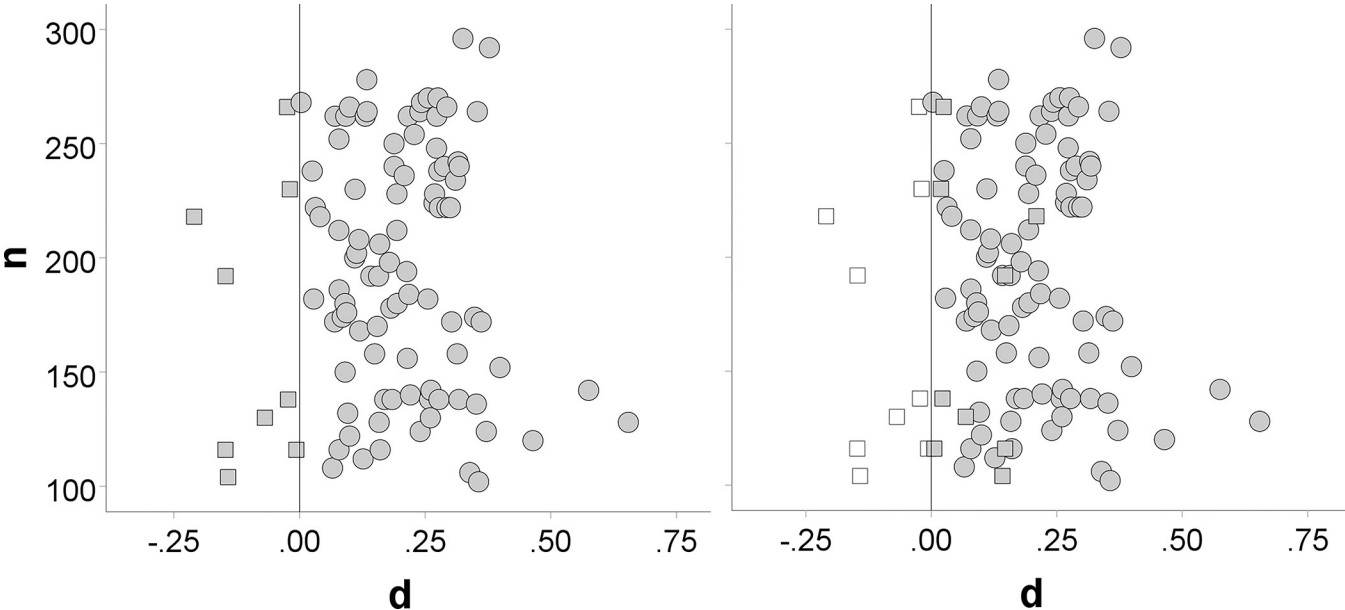

**Fig 2. Fictitious results for a set of 100 multiple replications of the same study.** Reflecting the absence of PSB, the n-ES correlation is $r = .00$ in the left panel (whereby squares/circles depict negative/positive effect sizes). However, the n-ES correlation is typically based on unsigned effect sizes [18]. Once these are used (filled elements in right panel, whereby squares indicate results with changed sign), the n-ES correlation changes to $r = -.07$.

large samples for topics that typically produce weak effects. As we discussed earlier, this could explain the negative n-ES correlation in cross-topics samples [18], even in the absence of PSB. The average effect size in a meta-analysis reflects the typical strengths of effect sizes for the topic under investigation. We therefore correlated meta-analyses' average effect size with their average sample size. If this n-ES correlation *between meta-analyses* was negative, this would indicate sample-size adjustment across topics. In this case, the n-ES correlation can be expected to be stronger in cross-topics samples than within meta-analyses, because sample-size adjustment is implausible in the latter (see also Table 2). To explore this idea was the third aim of Study 1. Finally, this study provided an opportunity to further examine the distribution of empirical effect sizes. A previous study [23] evaluated 12,170 correlation coefficients and 6,447 Cohen's *d* statistics extracted from studies included in 134 published meta-analyses. In their terminology the 25th, 50th and 75th percentiles are labelled small, medium, large and they contrasted these with Cohen's guidelines [24,25]. A survey [23] found that these empirical values were considerably lower than Cohen's guidelines (*d* = 0.15/0.36/0.65 instead of 0.20/0.50/ 0.80 for small, medium, and large effect sizes, respectively). In a sample of 150 meta-analyses, we compare our empirical estimates for small, medium, and large effect sizes to previous findings [23].

## Study 1

### Methods

**Samples.**   Our analyses require meta-analyses that report sample sizes and effect sizes for their primary studies. We used a compilation of such meta-analyses, 50 each for cognitive psychology, organizational psychology, and social psychology [15]. From the same source, we took 57 multiple replications as a comparison standard [21,22,26–30]. Following [18], the signed effect sizes were recoded as unsigned Cohen's *d*. The datasets are described in full in [15].

### Data analysis

To obtain results within meta-analyses and within multiple-replications results, we computed the n-ES correlation as Pearson's *r* for each of the meta-analyses and for each set of multiple replications. To facilitate comparisons with [18], we also calculated Spearman $\rho$ ($r_S$). Where relevant we used bootstrapping for comparing groups (10,000 bootstraps) [31]. All analyses were conducted in R 4.2.1 [32]. The data and analysis document (including additional analyses and robustness checks) can be found at https://osf.io/ce6v3/?view_only= 86b6b997ca52430898a6a2bdb38cf9bb.

### Results

**Small, medium, large effect sizes based on replication studies and meta-analyses.**   Following [23], we examined the 25th, 50th and 75th percentiles and labelled these small, medium, large effect sizes (Cohen *d*). For multiple replications, the values corresponding to small, medium, and large effect sizes were 0.12, 0.33 and 0.86, respectively.

For all meta-analyses, values for small, medium, large effect sizes were 0.18, 0.42 and 0.77. Dividing these by discipline, showed some minor variations. For cognitive psychology, the corresponding values were 0.24, 0.50 and 0.90. For organizational psychology, the corresponding values were comparable to those of cognitive psychology: 0.22, 0.45 and 0.80. For social psychology, the corresponding values were notably smaller than the two other disciplines: 0.13, 0.33 and 0.64.

**Table 1. Descriptive statistics for the correlation between sample size and effect size within meta-analyses (MAs) and multiple replications.** 95% CI based on bootstrap.

| Sample | Skew | M | SD | Mdn [95% CI] |
|---|---|---|---|---|
| $r$ | | | | |
| All MAs | 0.12 | -.13 | .24 | -.15 [-.11, -.18] |
| Cognitive psychology MAs | 1.10 | -.16 | .28 | -.20 [-.13, -.25] |
| Organizational psychology MAs | 0.46 | -.12 | .18 | -.14 [-.10, -.17] |
| Social psychology MAs | -1.01 | -.11 | .27 | -.11 [-.03, -.18] |
| Multiple replications | 0.26 | -.16 | .17 | -.18 [-.11, -.21] |
| $r_S$ | 0.12 | -.17 | .26 | -.16 [-.11, -.22] |
| All MAs | 0.83 | -.20 | .29 | -.22 [-.15, -.31] |
| Cognitive psychology MAs | -0.10 | -.12 | .21 | -.11 [-.03, -.17] |
| Organizational psychology MAs | -0.31 | -.18 | .29 | -.18 [-.10, -.27] |
| Social psychology MAs | -0.26 | -.15 | .24 | -.16 [-.07, -.19] |
| Multiple replications | | | | |

**The n-ES correlation in the absence of sample-size adjustment: Within meta-analyses and multiple replications.** Addressing our first aim, we first focus on the n-ES correlation within meta-analyses and multiple replications. As discussed earlier, both should be unaffected by sample-size adjustment. Additionally, multiple replications are also unaffected by PSB (see also Table 2). Descriptive statistics are presented in Table 1.

As can be seen, $r$ and $r_S$ produced very similar results for the n-ES correlation within meta-analyses (Table 1). Likewise, whether the average correlation was expressed as mean or median hardly affected results. In the remainder, we follow [18] and focus on $r_S$; for consistency with subsequent analyses, we describe the average n-ES correlation via the median. Consistently throughout domains, negative n-ES correlations emerged, with averages ranging from small to small-to-medium in strength. All median n-ES correlations differed statistically significantly from zero (because the confidence intervals excluded zero, see Table 1). Interestingly, we found the same n-ES correlation within meta-analyses and multiple replications, median $r_S$ = -.16 (see also Table 2, which summarises key results). As discussed earlier, this similarity would be expected if meta-analyses are unaffected from PSB.

As discussed earlier, the negative n-ES correlation likely arises from our reliance on unsigned effect sizes (see Fig 2). In order to test this explanation, we re-ran the n-ES

**Table 2. Study types, their characteristics, and key results.**

| Study type | Sample-size adjustment possible? | Publication selection bias possible? | Effect size type | $r_{n\text{-}ES}$ |
|---|---|---|---|---|
| *Study 1* | | | | |
| within meta-analyses | no | yes | focal + periph. | -.16 |
| multiple replications | no | no | –[b] | -.16 |
| between meta-analyses | yes | –[a] | –[a] | -.24 |
| cross-topics (from meta-analyses) | yes | yes | focal + periph. | -.23 |
| [18] | yes | yes | focal | -.45 |
| cross-topics | no | yes | focal + periph. | -.16[c] |
| [33] | yes | yes | focal | -.55 |
| within meta-analyses | yes | yes | focal + periph. | -.37 |
| *Study 2* | | | | |
| cross-topics, focal effect size | | | | |
| cross-topics, random effect size | | | | |

Notes

[a]The analysis correlates meta-analyses' average effect size with their average sample size. At this level of aggregation, these aspects are immaterial.

[b]Effect sizes in multiple replications focus on hypotheses; however, this is immaterial here because publication selection bias can be ruled out.

[c]Pearson's $r$ (all other correlations are $r_S$).

correlations within multiple replications. This time, however, we used signed effect sizes within each set of multiple replications (akin to the left panel in Fig 2). In line with our explanation, the median n-ES correlation fell to $r_S$ = -.01 [-.07, .08]. The median for the signed n-ES correlation within meta-analyses was $r_S$ = -.04 [-.10, -.0003]. Albeit statistically significant, because the 95% CI excludes zero, this correlation is very small.

## Exploring sample-size adjustment: The n-ES correlation between meta-analyses

Addressing our second aim, we investigated sample-size adjustment and checked if studies on topics that tend to produce relatively large effect sizes tend to have relatively small $n$. We therefore examined the correlation between meta-analyses' average $n$ and average effect size. Within meta-analyses, average $n$ tended to be strongly right-skewed ($Mdn_{skewness}$ = 2.38); average effect size also tended to be right-skewed, but to a lesser extent ($Mdn_{skewness}$ = 1.05). For each meta-analysis, we therefore expressed its average effect size via the mean and its average $n$ via both its mean and its median. We then ran two sets of analyses, one based on mean $n$, and one based on median $n$. Both led to very similar results and identical conclusions. Here, we report the analyses based on median $n$.

The scatterplot for the relationship between meta-analyses' average $n$ and average effect size showed strong outliers (see Fig 3). Consequently, we focussed on $r_S$, which resulted in a small-to-medium negative correlation ($r_S$ = -.24, $p$ = .003). (Note that the same correlation was statistically nonsignificant when expressed as $r$; $r$ = -.14, $p$ = .082.) As discussed earlier, this pattern is indicative of sample-size adjustment across topics.

**Comparing the n-ES correlation within meta-analyses and cross-topics.** Irrespective of PSB, sample size adjustment (which is plausible cross topics but not within meta-analyses) should lead to a higher n-ES correlation in cross-topics samples than within meta-analyses. Our previous analysis investigated the n-ES correlation cross topics, but at a high level of aggregation (meta-analyses' average effect size and average $n$). This precludes a sensible comparison with our earlier results regarding the n-ES correlation within meta-analyses, which was investigated at a more granular study level. To enable such a comparison, we pooled all primary studies across our 150 meta-analyses, treated them as a single cross-topics sample, and computed a single n-ES correlation as [18] did.

The 150 meta-analyses comprised altogether 7,227 primary effect sizes and sample sizes. Right-skew was observed for $d$ (4.3) and particularly for $n$ (78.2). Medians ($M$, $SD$) were 0.42 (0.57, 0.62) and 100 (438, 8131), respectively. The n-ES correlation across topics was $r_S$ = -.23, 95% CI [-.21, -.25], only slightly higher than our average n-ES correlation within meta-analyses (median $r_S$ = -.16). This suggests that the effect of sample-size adjustment on the n-ES correlation in cross-topics samples is modest.

A previous study [18] found a much stronger n-ES correlation in their cross-topics sample ($r_S$ = -.45) To facilitate comparisons, we computed an estimated 95% CI [34], which was [-.36, -.53]. This differs markedly from the CI for our n-ES correlation across topics [-.21, -.25]. Consequently, sampling error cannot easily account for the stark difference in the n-ES correlation in our cross-topics sample and in [18].

## Discussion

In a sample of 150 meta-analyses, we found, on average, a fairly small negative n-ES correlation (mean $r$ = -.13). This is virtually the same as the mean n-ES correlation of $r$ = -.16 previously observed, with the same methods, in another sample of 75 psychology meta-analyses [33]. The authors interpreted their result as evidence that meta-analyses are frequently affected by

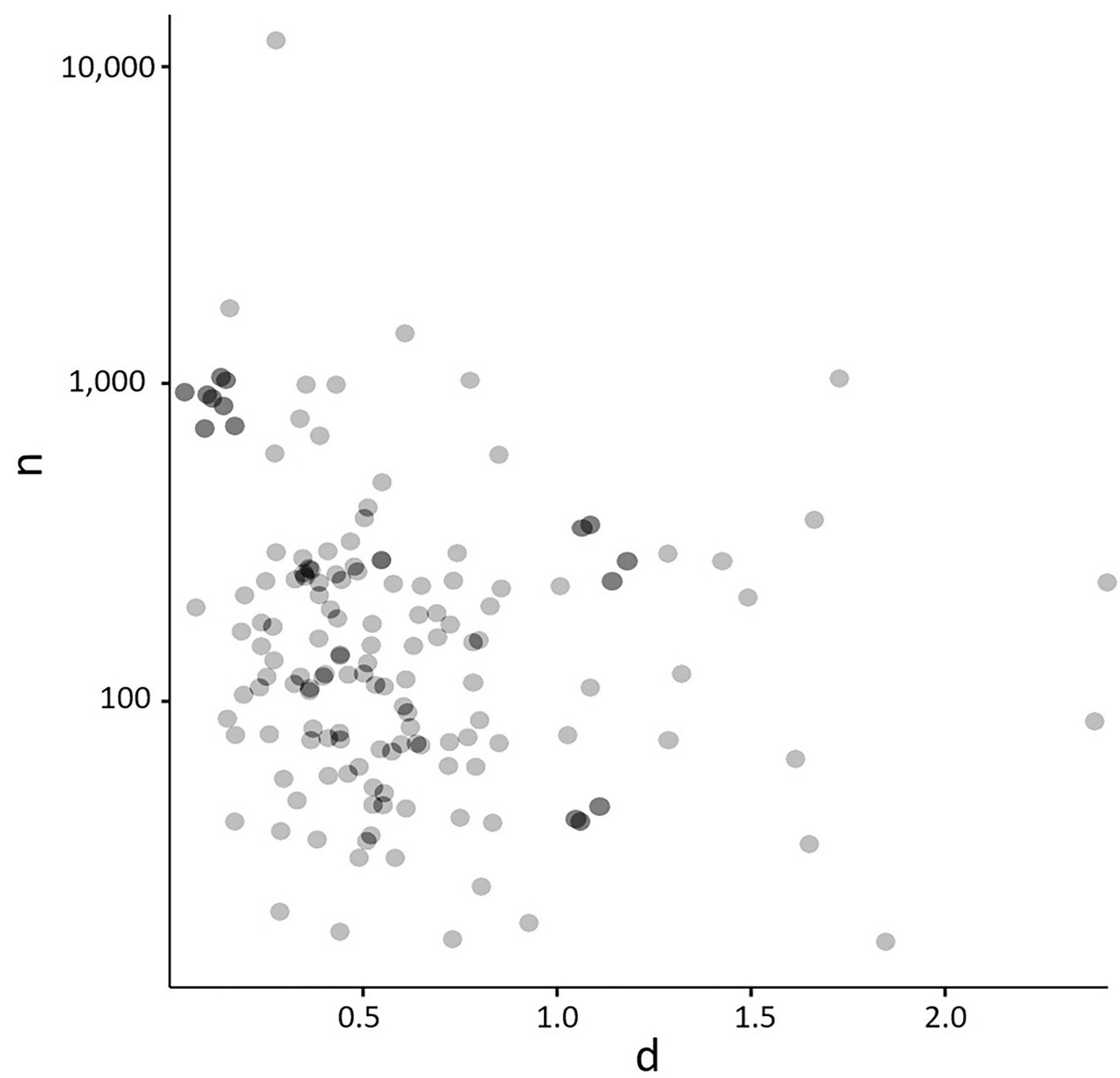

**Fig 3. Relationship between meta-analyses' average effect size (absolute Cohen's *d*) and their average sample size (*n*, on a log scale).**

publication bias. Our results offer a different perspective. We found the same negative n-ES correlation in multiple replications, which are free from publication bias (and PSB, more generally), and we showed that the negative n-ES correlation is mostly a statistical artifact that arises from using unsigned effect sizes. Therefore, findings regarding n-ES correlations within meta-analyses offer, in our view, little evidence for PSB in psychology meta-analyses.

Previously, a much stronger n-ES correlation ($r_S$ = -.45) was observed in a cross-topics sample [18]. These authors, too, interpreted their finding as evidence for pervasive publication bias. As we argued in the introduction, their n-ES correlation might reflect (problematic) PSB,

(innocuous) sample-size adjustment, or both. Our analyses found evidence for sample-size adjustment; studies that investigated stronger effects (as indicated by the overall meta-analytic effect size) tended to rely on smaller sample sizes than studies that investigated weaker effects. However, sample-size adjustment cannot fully explain the gap in the n-ES correlation within meta-analyses vs. across topics: When we combined all meta-analyses into one large cross-topics sample, our n-ES correlation ($r_S$ = -.23) remained much smaller than reported previously (see also Table 2).

Why might this be the case? The previous cross-topics sample took effect sizes from findings that directly addressed the main research question of the respective publication [18]. In contrast, meta-analyses include *any* pertinent result, regardless of whether it was focal or peripheral to the study it emerges from. Plausibly, PSB might be stronger for results that are focal to a study and weaker or absent for results that are peripheral. This could explain the difference between the small n-ES correlation in our cross-topics analysis of meta-analyses and the previous cross-topics sample of focal findings.

## Study 2

The aim of Study 2 was therefore to compare the n-ES correlation between focal effect sizes (i.e., those that address the study's central hypothesis or aim) and random effect sizes in a cross-topics sample.

As we explain in this section, such a comparison should take the design of the study (between- versus within-subjects) into account. In a within-subjects design, there are two ways to translate the difference between two means into a standardised effect size (e.g., [35]). This difference can be standardised with the pooled standard deviation across the two conditions. This is the same type of effect size that arises from between-subject designs (henceforth, $ES_{between}$). Alternatively, the difference between means can be standardised with the standard deviation for participants' change scores; this approach typically results in a larger effect size (henceforth, $ES_{within}$). Especially when participants' scores correlate strongly across conditions (e.g., because the treatment effect is very homogenous across participants), $ES_{within}$ can be much larger than $ES_{between}$.

In surveys that investigate the n-ES correlation, effect sizes from within-subjects designs will often be of the $ES_{within}$ type because information to compute $ES_{between}$ is lacking from the primary study. (If effect sizes are taken from a meta-analysis, its authors might have chosen to compute $ES_{within}$.) At the same time, within-subjects designs have greater statistical power than between-subjects designs, leading researchers to choose a relatively small *n*. Consequently, it can be expected that $ES_{within}$, compared against $ES_{between}$, tends to be both large and associated with small *n*. This would, similar to Simpson's paradox [36], negatively bias the n-ES correlation without being indicative of PSB (see Fig 4). For this reason, it is worthwhile to take the design of the study into account.

### Method

**Power analysis.** We sought 90% power to identify a difference between two dependent correlations, *r* = -.45 and *r* = -.16, via a two-tailed test with α = .05. Our power analysis in G*Power [37] suggested a minimum sample size of *n* = 157. We decided to use a sample of 160 papers. (The Open Science Framework page for this paper contains alternative power analyses with varying assumptions. In all cases, *n* ~ 150 appeared sensible for correlations with dependency, https://osf.io/ce6v3/?view_only=86b6b997ca52430898a6a2bdb38cf9bb.)

**Eligibility criteria and sampling of papers.** To be suitable for our study, psychology papers needed to fulfil the following eligibility criteria: present original data; use inferential

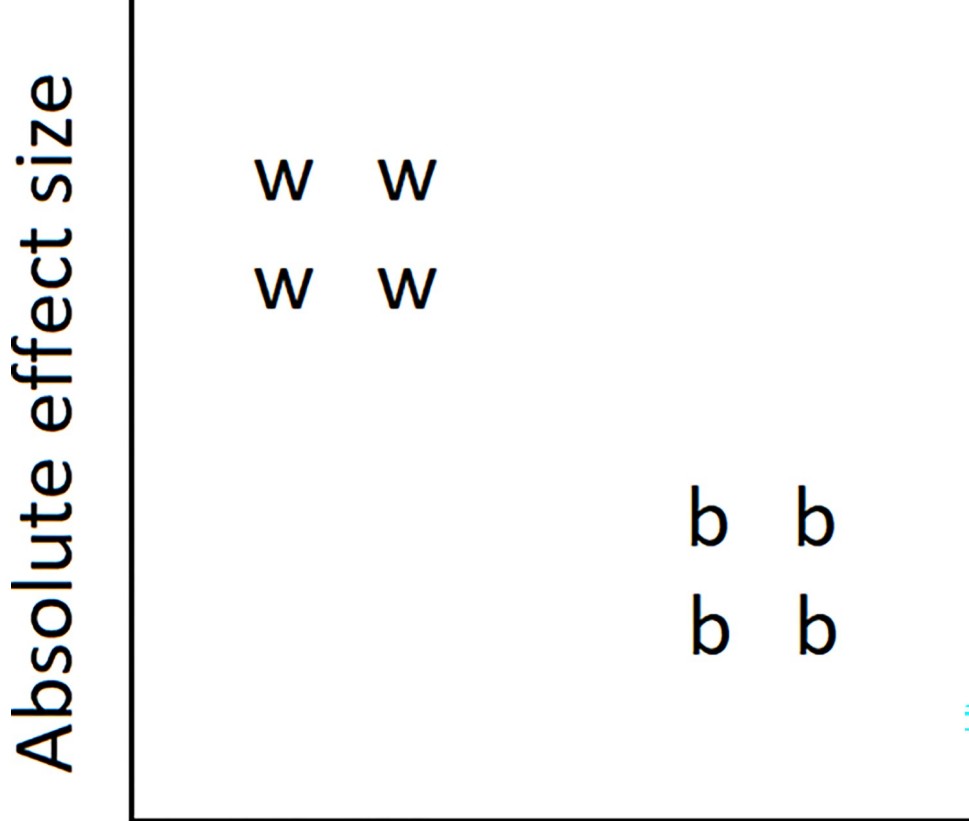

**Fig 4. Funnel plot for hypothetical results from four within-subjects studies (ws) and four between-subjects studies (bs).** Overall, a strong n-ES correlation emerges, although this correlation is zero for both types of study design.

statistics to address the main research question; provide sufficient information to calculate relevant effect sizes; present $n$. We excluded papers that focused on inferential analyses for which there is no straightforward unitary effect size (multilevel models, structural equation models, time series models, cluster analysis, social network analysis, multidimensional scaling, statistical simulation models, machine learning models, exploratory factor analysis and principal component analysis).

To sample 160 papers, we (somewhat arbitrarily) decided to draw 16 papers for each year from 2012–2021. In particular, we searched for "the" in All Fields in *Web of Science*, restricted by target year. To focus on psychology papers, we used Web of Science Category and retained only those categories that start with "psychology". From the resulting list of hits, we selected a paper with the help of a random number generator. If the paper fulfilled our eligibility criteria, it was retained; otherwise, we moved down the list until a suitable paper was found. This process was repeated with new random numbers until all 16 papers for that year were retained. The same process was then repeated for all years.

**Selection and coding of focal and random effect size.** For each paper, we extracted two effect sizes, one focal and one random, as well as the sample size associated with each. The

**Table 3. Descriptive statistics for focal and random effect sizes and their associated sample sizes in Study 2.**

| | n | Effect size | | | Sample size | | |
|---|---|---|---|---|---|---|---|
| | | Skew | M(SD) | Mdn | Skew | M(SD) | Mdn |
| *Focal* | | | | | | | |
| all studies | 160 | 2.21 | 0.84(0.82) | 0.63 | 9.27 | 1396(9252) | 84 |
| between-subjects | 135 | 3.08 | 0.68(0.68) | 0.54 | 8.48 | 1645(10058) | 97 |
| within-subjects | 25 | 0.81 | 1.72(1.00) | 1.51 | 3.08 | 49(70) | 25 |
| *Random* | | | | | | | |
| all studies | 160 | 2.69 | 0.83(1.02) | 0.52 | 8.16 | 1058(5423) | 82 |
| between-subjects | 135 | 3.30 | 0.72(0.91) | 0.46 | 7.46 | 1244(5888) | 103 |
| within-subjects | 25 | 1.21 | 1.46(1.35) | 0.84 | 3.63 | 55(93) | 26 |

focal effect size directly addressed what the paper presented as its main hypothesis or aim. If the paper presented multiple hypotheses/aims as equally important, we used the one mentioned first in its hypotheses/aims section. One author (JH) identified the focal aim/hypothesis for all papers without knowledge of their analyses and results. In cases in which the paper later proved to have no effect size information for this aim/hypothesis, we moved to its next aim/hypothesis. Where multiple outcome variables, samples, or analyses were relevant for the focal effect size, we used whichever occurred first (either in a table or in text) in the results section.

In each paper, we choose a second effect size at random. (By chance, this sometimes happened to be the same as the focal effect size.) We selected a page via a random number generator. We coded the first effect size information on that page that originated from the paper's study. For this purpose, we read any tables line-by-line, not column-by-column. If the page did not contain relevant effect size information, we repeated the process as required.

All effect sizes were coded as unsigned $d$. We used various online calculators to convert descriptive statistics, effect sizes (e.g., $\eta^2$, $R^2$, $r$, $r_S$, odds ratios), and various test statistics (e.g., $F$-value, $t$-value, $\chi^2$) into $d$. Details on extraction and conversion are provided in our pre-registration document on the OSF.

All coding of effect sizes and sample sizes was done by one author (JH). To check reliability, we selected 40 papers at random. Based on the identified focal aim/hypothesis, a second author (AL or TP) independently coded the focal effect size and its associated sample size. Again, both $d$ and $n$ proved strongly right skewed (see Table 3), which is why we computed $r_S$. Correlations between first and second coder proved satisfactory, with $r_S$ = .74 for $d$ and $r_S$ = .97 for $n$.

## Analytical strategy

The design of the study was pre-registered, and the analyses were conducted in R 4.2.1 [38]. We preregistered the comparison of the n-ES correlation between focal and randomly selected effects based on Pearson $r$. For this, we used Zou's method [39], which is based on the (non-) overlap of confidence intervals and allows for dependency between correlations. Analyses were performed with cocor [40]. We used 89% confidence intervals here [41]. As visual checks showed that relevant distributions were distinctly non-normal, we deviated from the preregistration and relied on $r_S$ rather than Pearson's $r$. (The OSF contains additional analyses, with 95% CI such as those based on Percentage Bend correlation leading the same conclusion. https://osf.io/ce6v3/?view_only=86b6b997ca52430898a6a2bdb38cf9bb)

## Results and discussion

Descriptive statistics are shown in Table 3. For focal findings, we found a very strong negative n-ES correlation, $r_S$ = -.55, 89% CI [-.64, -.45]. In line with our reasoning, this correlation

turned out to be smaller for randomly selected effect sizes, $r_S$ = -.37, 89% CI [-.48, -.22]. However, the 89% confidence intervals overlapped, and we therefore conclude that these correlations do not offer convincing support for our hypothesis that the n-ES correlation is stronger for focal effects than for effects chosen at random. This conclusion was not altered when we performed the analyses by type of design (between- vs. within-subjects). For between-subject designs ($n$ = 135), we found a strong negative n-ES correlation for focal effect sizes $r_S$ = -.40, 89% CI [-.52, -.27] and a smaller one for randomly selected effect sizes, $r_S$ = -.29, 89% CI [-.42, -.16]. For within-subject designs ($n$ = 25), we found a very strong negative n-ES correlation for focal effect sizes $r_S$ = -.67, 89% CI [-.83, -.41] and a much smaller one for randomly selected effect sizes, $r_S$ = -.17, 89% CI [-.48, .17].

Our analysis of focal findings largely followed the methods in [18]. We note that, in contrast to Study 1, our result ($r_S$ = -.55) was now quite similar to theirs ($r_S$ = -.45).

Further, comparisons of results across Studies 1 and 2 are instructive. The n-ES correlations in Study 2's focal effects [-.64, -.45] and in Study 1's cross-topics analysis across meta-analyses [-.25, -.22] differed reliably. This confirms our conclusion from Study 1 that the n-ES correlation is much stronger for effects sampled from the effects publications focus on than for effects sampled from meta-analyses. Further, the n-ES correlations in Study 2's randomly selected effects [-.48, -.22] and in Study 1's cross-topics analysis across meta-analyses [-.25, -.22] failed to differ reliably. Thus, the n-ES correlation for randomly selected effects is therefore not clearly more worrying than for focal effects nor clearly less worrying than for effects in meta-analyses. In light of these inconclusive results, we struggle to understand why the n-ES correlation differs so dramatically between effects in meta-analyses and publications' focal effects. We note that meta-analyses in Study 1 stemmed from only three sub-disciplines whereas samples of focal effects stemmed from all of psychology, but it remains currently unclear if this can explain the observed differences.

## General discussion

The n-ES correlation holds promise to indicate how widespread a problem PSB is and, following the introduction of counter measures, how this might change over time [14,18,33]). However, the n-ES correlation is also affected by researchers' (unwitting or deliberate) adjustments of their sample size to the expected effect size, a perfectly reasonable behavior. Using data from psychology, we therefore investigated in greater detail to what extent the n-ES correlation suggests the presence of PSB in psychological research.

In Study 1, we found a small negative n-ES correlation within meta-analyses, which is consistent with previous results [33]. This proved to be virtually identical with the negative n-ES correlation that we observed in multiple replications, which are free from PSB. We also showed that a small negative n-ES correlations like these are plausible in the absence of PSB. Overall, we would therefore argue that the small negative n-ES correlation within psychological meta-analyses consistently observed by us, and by [33] suggests the absence of noteworthy PSB (at least in the three scrutinized sub-disciplines cognitive, organizational, and social psychology). (Similarly, our results suggest an n-ES correlation around $r_s$ = -.23 is no reason for concern.) This is in line with previous research which suggests that evidence for PSB in psychological meta-analyses is weak, and if PSB is present it is likely to be mild [17]. Similarly, previous research indicates that applying adjustments for PSB to psychological meta-analyses results in minimal changes to effect size estimates [42]. Obviously, that does not mean that PSB is never a problem in meta-analyses in psychology, and research into how best to uncover it remains important (e.g., [5,16,43].

The inconspicuous n-ES correlation for effects sampled from meta-analyses contrasts sharply with the one in cross-topics samples of focal effects (i.e., effects that take a central role in the papers they are published in): For cross-topics samples of focal effects, [18] and our Study 2 consistently found strong negative n-ES correlations. At a theoretical level, such a difference might be expected. First, across different topics researchers might (unwittingly or deliberately) adjust their sample size to the expected effect size, which induces a negative n-ES correlation. Such sample-size adjustment is less plausible to occur within meta-analyses. Here, researchers investigate the same topic and therefore would rarely have reasons to hold different expectations about the magnitude of the expected effect [15]. Second, it is plausible that PSB should affect focal effects in particular. For example, researchers who fail to find an expected effect but find an unexpected one instead might shift the focus of their publication on the latter [44]. By definition, cross-topics samples of focal effects consist of focal effects only, which is not true for meta-analyses. Thus, a smaller proportion of effects in meta-analyses should be affected by PSB, thus reducing the n-ES correlations within meta-analyses.

Our empirical evidence did not suggest that sample-size adjustment and stronger PSB in focal effects sufficiently account for the large difference in the n-ES correlation within meta-analyses versus cross-topics samples of focal effects. Although, we found evidence for sample-size adjustment in Study 1, this was too weak to explain the difference in the n-ES correlation within meta-analyses and across topics. Moreover, we failed to find clear evidence in Study 2 that the n-ES correlation is less pronounced for effects selected at random than for focal effects. In sum, it remains currently unclear why much stronger n-ES correlations are found in samples of focal effects than in samples of effects in meta-analyses and to what extent this reflects benign or problematic reasons. Although, our research suggests that some negative n-ES correlations might be seen as unproblematic, it currently remains unclear how strong n-ES correlations need to be to indicate nontrivial PSB effects. More research on these topics is needed.

The n-ES correlation is one among numerous indicators developed to indicate the presence of PSB (e.g., [16]). Here, we focussed on the n-ES correlation because previous surveys based on this method have fuelled concerns about widespread PSB in psychological research [18,33]. Various methods have been compared regarding their ability to uncover PSB in single meta-analyses (e.g., [16,17]. Whether the n-ES and other indicators differ in their suitability to describe PSB in the kind of larger surveys that we presented here is currently unclear.

## Conclusion

Negative n-ES correlations have previously been described as evidence for PSB in psychological research [18,33]. We demonstrated here that the negative n-ES correlations in meta-analyses from three psychological sub-disciplines were inconspicuous and do not point to worrying levels of PSB. However, alternative sampling strategies lead to much stronger n-ES correlations partly explained by benign factors. The extent to which these heightened n-ES correlations also reflect the effects of PSB is currently unclear.

## Author Contributions

**Conceptualization:** Thomas V. Pollet, Johannes Hönekopp.

**Data curation:** Thomas V. Pollet.

**Formal analysis:** Thomas V. Pollet.

**Investigation:** Audrey Helen Linden, Johannes Hönekopp.

**Methodology:** Audrey Helen Linden, Thomas V. Pollet, Johannes Hönekopp.

**Writing – original draft:** Johannes Hönekopp.

**Writing – review & editing:** Audrey Helen Linden, Thomas V. Pollet.

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
