## [Decision Letter · Decision Letter 0]

5 Nov 2023

PONE-D-23-22887Publication Bias in Psychology: A Closer Look at the Correlation Between Sample Size and Effect SizePLOS ONE

Dear Dr. Hönekopp,

Thank you for submitting your manuscript to PLOS ONE. After careful consideration, we feel that it has merit but does not fully meet PLOS ONE’s publication criteria as it currently stands. Therefore, we invite you to submit a revised version of the manuscript that addresses the points raised during the review process.

I agree with the concepts of both reviewers. The manuscript in its current form needs a major revision.

We look forward to receiving your revised manuscript.

Kind regards,

Diego A. Forero, MD; PhD

Academic Editor

PLOS ONE

2. Please upload a copy of Figure 1, to which you refer in your text on page 5. If the figure is no longer to be included as part of the submission please remove all reference to it within the text.

Additional Editor Comments:

I agree with the concepts of both reviewers. The manuscript in its current form needs a major revision.

Reviewers' comments:

Reviewer's Responses to Questions

**Comments to the Author**

1. Is the manuscript technically sound, and do the data support the conclusions?

Reviewer #1: Yes

Reviewer #2: No

2. Has the statistical analysis been performed appropriately and rigorously? 

Reviewer #1: Yes

Reviewer #2: No

3. Have the authors made all data underlying the findings in their manuscript fully available?

Reviewer #1: Yes

Reviewer #2: Yes

4. Is the manuscript presented in an intelligible fashion and written in standard English?

Reviewer #1: Yes

Reviewer #2: Yes

5. Review Comments to the Author

Reviewer #1: The paper delves into the observed negative correlation between sample size and effect size in published reports within the field of psychology. It conducts two comprehensive studies to ascertain the underlying reasons for this correlation and to examine the impact of publication bias. In Study 1, the authors compare data from 150 meta-analyses across various subfields of psychology with 57 multiple replication datasets. They argue that because the replications are pre-registered, the chances of publication bias affecting them are lower. Based on their findings, the authors suggest that the small negative correlation between sample size and effect size is more likely the result of sampling adjustments, where researchers might be conducting smaller studies to capture larger effect sizes. Study 2 further scrutinizes this phenomenon by examining 160 randomly selected psychology papers. The authors find that focal effects (those central to the study’s hypothesis) tend to have larger negative correlations between sample size and effect size compared to randomly selected effects. However, these differences were not statistically significant. This leads to inconclusive evidence on whether publication bias has a differential impact on focal versus random effects. Furthermore, the negative correlation for random effects did not differ substantially from that found in meta-analyses, suggesting that publication bias might not be more pervasive in one versus the other. While the paper clarifies some aspects of the relationship between sample size and effect size, it leaves unanswered questions regarding the role of publication bias, particularly in focal versus random effects. The authors, recognizing these gaps, call for further research to better understand these nuances.

I find the authors’ approach to be commendable and thorough. The paper certainly contributes to our understanding of a complex issue, yet I find that the conclusions offer more questions than answers.

I also have concerns about the clarity of the paper, specifically with respect to the description of Study 1. The rationale for the different comparisons conducted is not clearly outlined, leaving me confused about which comparisons were pre-planned and which were post hoc. This lack of clarity makes it difficult to fully grasp the methodology and logic behind each comparison. For instance, the meaning of “cross-topics samples” and “within meta-analyses” does not become clear until much later in the paper from where these terms are first mentioned. The rational for the analyses is finally revealed in the discussion where the authors suddenly bring an important conclusion that there is little evidence of publication bias in psychology meta-analyses. It would have been much better to pre-empt this conclusion with an explanation in the methods detailing which results might support such a conclusion. Additionally, at the outset of Study 1, the reason for selecting meta-analyses is stated as the absence of sample size adjustments. Yet this seems to be the exact explanation the authors used to justify the small negative correlation found in Study 1. Please include Figure 1 in the next submission as it is missing from the manuscript.

Additionally, I noticed that in Study 2, the agreement for converting effect sizes to Cohen's d between the two raters is surprisingly low (r=0.74). Given that this is supposed to be a formulaic calculation, this low level of agreement should have been discussed as a limitation in the paper.

In the general discussion section, the authors should discuss whether the correlation between effect size and sample size is indeed the most appropriate measure to use for this investigation. The authors advocate for this metric, but they do not thoroughly explore its limitations or consider alternative metrics that could also be informative.

In summary, while I appreciate the paper's aim and the rigor of its investigation, I believe it needs substantial revisions for clarity, discussion of limitations, and a more robust discussion of its key metric before it is ready for publication.

Reviewer #2: Review of Linden et al., ‘Publication bias in psychology: a closer look at the correlation between sample size and effect size’, submitted to PLoS ONE.

Summary

This study uses data from meta analyses and replication studies to investigate the correlation between sample size and effect size, which is typically negative. The results are obscured by a statistical artifact that is a consequence of taking absolute effect sizes, and are therefore difficult to interpret. It seems that this artefact is a consequence of using methods from previous studies, so I think there is potential for this paper to be reimagined as a critique of these methods, using the existing data and simulations. There are also some sloppy errors throughout, especially surrounding figures.

Specific points

1. The most serious issue I have with this work is that using the absolute effect size is almost guaranteed to introduce a spurious correlation in the direction reported. Since the authors use R, the problem can be illustrated by the following code, which simulates a series of random effect sizes and sample sizes centred on a true effect size of 0. By taking the absolute effect sizes, we introduce a strong negative correlation:

par(mfrow=c(1,2), las=1)

nstudies <- 200 # choose how many studies to include

trued <- 0 # set the true underlying effect size

truesd <- 1 # set the standard deviation of the underlying effects

alld <- NULL

allN <- NULL

# simulate some studies using random sample sizes and effect sizes

for (n in 1:nstudies){

allN[n] <- 3+round(50*abs(rnorm(1)))

alld[n] <- mean(rnorm(allN[n],mean=trued,sd=truesd))

}

# plot the signed effect sizes and calculate regression slope

plot(c(trued,trued),c(0,100),type='l',lwd=2,xlim=c(trued-truesd,trued+truesd),ylim=c(0,100),ann=FALSE,col=rgb(0.5,0.5,0.5))

points(alld,allN,pch=16,cex=0.5)

modout <- lm(alld ~ allN)

lines(modout$coefficients[1] + modout$coefficients[2]*(0:100),0:100,lwd=3,col='red')

title(xlab="Effect size (d)", col.lab=rgb(0,0,0), line=2.2, cex.lab=1.5) # titles for axes

title(ylab="Sample size (N)", col.lab=rgb(0,0,0), line=2.2, cex.lab=1.5)

title(main="Signed effect sizes", cex.main=1.5)

# plot the unsigned effect sizes and calculate regression slope

plot(c(trued,trued),c(0,100),type='l',lwd=2,xlim=c(trued-truesd,trued+truesd),ylim=c(0,100),ann=FALSE,col=rgb(0.5,0.5,0.5))

points(abs(alld),allN,pch=16,cex=0.5)

modout <- lm(abs(alld) ~ allN)

lines(modout$coefficients[1] + modout$coefficients[2]*(0:100),0:100,lwd=3,col='red')

title(xlab="Effect size (d)", col.lab=rgb(0,0,0), line=2.2, cex.lab=1.5) # titles for axes

title(ylab="Sample size (N)", col.lab=rgb(0,0,0), line=2.2, cex.lab=1.5)

title(main="Unsigned effect sizes", cex.main=1.5)

The spurious correlation will become smaller as the mean effect departs from 0, which might explain some of the differences (i.e. between meta-analyses and replication studies) reported in the manuscript. It’s not clear to me if the authors are fully aware of this problem, even though it is very clear in Figure 2. There is a single mention of a statistical artifact on page 13, but it’s not explained in detail so I can’t tell if this is what’s being referred to or if it’s something else. I suspect this error is not the fault of the authors – it sounds like the convention of taking the absolute value has come from earlier studies, but it is clearly a huge problem. I would recommend repeating all analyses using the signed effect sizes, and using this comparison (as in Fig 2) as a critique of calculating the n-ES correlation using absolute values.

2. Figure 1 is completely missing from the manuscript. On page 5 it says ‘see Fig. 1 [to be added]’.

3. Plots of effect size against sample size are usually referred to as funnel plots. It’s worth using this terminology so that readers familiar with funnel plots realise that you’re talking about the same thing. Also, funnel plots are usually plotted with the effect size on the x-axis and the sample size on the y-axis. Is there a reason not to use this convention here (i.e. in Figs 2-4)?

4. First paragraph – it’s not just PhD students that waste their time researching imaginary effects! I would change this to ‘and researchers might waste their time investigating’.

5. Top of page 4 – it’s not clear why the correlation between N and effect size is ‘more suitable’, so this needs clarifying.

6. Page 8 – the relevance of the sqrt(n) point about correlations is unclear and needs explaining.

7. I don’t buy the argument that meta-analyses are immune to sample-size adjustment. It’s usually the case that sample size estimates are conducted using effect sizes from previous relevant literature, so this must surely involve an influence of earlier studies on later ones. For example, if the first study reporting an effect has a huge effect size, the sample size of later studies will be informed by this (and will probably be small).

8. Figure 2 is described backwards. The left panel is the abs values, but the caption says it’s the right panel. I think the two panels are just the wrong way around.

9. Figure 4 is also described incorrectly. The text describes the within subjects studies as being small in sample size and large in effect size, which corresponds to the black crosses, but the figure caption says the within-subjects studies are the grey crosses. Also the two grey levels are quite hard to distinguish – consider using a different symbol for one condition.

6. PLOS authors have the option to publish the peer review history of their article (what does this mean?). If published, this will include your full peer review and any attached files.

Reviewer #1: **Yes: **Olga Boukrina

Reviewer #2: No

---

## [Decision Letter · Decision Letter 1]

28 Dec 2023

Publication Bias in Psychology: A Closer Look at the Correlation Between Sample Size and Effect Size

PONE-D-23-22887R1

Dear Dr. Hönekopp,

We’re pleased to inform you that your manuscript has been judged scientifically suitable for publication and will be formally accepted for publication once it meets all outstanding technical requirements.

Kind regards,

Diego A. Forero, MD; PhD

Academic Editor

PLOS ONE

Additional Editor Comments (optional):

I agree with the two reviewers regarding that the authors have incorporated the previous suggestions into the revised manuscript.

Reviewers' comments:

Reviewer's Responses to Questions

**Comments to the Author**

1. If the authors have adequately addressed your comments raised in a previous round of review and you feel that this manuscript is now acceptable for publication, you may indicate that here to bypass the “Comments to the Author” section, enter your conflict of interest statement in the “Confidential to Editor” section, and submit your "Accept" recommendation.

Reviewer #1: All comments have been addressed

Reviewer #2: All comments have been addressed

2. Is the manuscript technically sound, and do the data support the conclusions?

Reviewer #1: Yes

Reviewer #2: Yes

3. Has the statistical analysis been performed appropriately and rigorously? 

Reviewer #1: Yes

Reviewer #2: Yes

4. Have the authors made all data underlying the findings in their manuscript fully available?

Reviewer #1: Yes

Reviewer #2: Yes

5. Is the manuscript presented in an intelligible fashion and written in standard English?

Reviewer #1: Yes

Reviewer #2: Yes

6. Review Comments to the Author

Reviewer #1: The authors have adequately addressed my concerns. For the final proof I recommend they correct the following typos:

Page 7 last paragraph, correct the grammatical error in phrase: “what to expected in the absence of PSB”

Page 12 last paragraph, “unaffected from PSB” should be “unaffected by PSB”

Page 24 1st paragraph, missing “to” in "leading the same conclusion”

Reviewer #2: (No Response)

7. PLOS authors have the option to publish the peer review history of their article (what does this mean?). If published, this will include your full peer review and any attached files.

Reviewer #1: **Yes: **Olga Boukrina

Reviewer #2: No

---

## [Editor Report · Acceptance letter]

7 Feb 2024

PONE-D-23-22887R1 

PLOS ONE

Dear Dr. Hönekopp, 

I'm pleased to inform you that your manuscript has been deemed suitable for publication in PLOS ONE. Congratulations! Your manuscript is now being handed over to our production team.

Kind regards, 

on behalf of

Dr. Diego A. Forero 

Academic Editor

PLOS ONE